# Cross-model Control: Improving Multiple Large Language Models in One-time Training

**Jiayi Wu[1], Hao Sun[2], Hengyi Cai[3], Lixin Su[4],**
**Shuaiqiang Wang[4], Dawei Yin[4], Xiang Li[1],[*] Ming Gao[1,5,6]**
[1]School of Data Science and Engineering, East China Normal University
[2]Peking University [3]Chinese Academy of Sciences [4]Baidu Inc
[5]KLATASDS-MOE, School of Statistics, East China Normal University
[6]Guizhou Zhuwen ECNU Data Power Institute
jiayiwu@stu.ecnu.edu.cn, sunhao@stu.pku.edu.cn
caihengyi@ict.ac.cn, {sulixin,wangshuaiqiang}@baidu.com
yindawei@acm.org, {xiangli,mgao}@dase.ecnu.edu.cn

## Abstract

The number of large language models (LLMs) with varying parameter scales and vocabularies is increasing. While they deliver powerful performance, they also face a set of common optimization needs to meet specific requirements or standards, such as instruction following or avoiding the output of sensitive information from the real world. However, how to reuse the fine-tuning outcomes of one model to other models to reduce training costs remains a challenge. To bridge this gap, we introduce Cross-model Control (CMC), a method that improves multiple LLMs in one-time training with a portable tiny language model. Specifically, we have observed that the logit shift before and after fine-tuning is remarkably similar across different models. Based on this insight, we incorporate a tiny language model with a minimal number of parameters. By training alongside a frozen template LLM, the tiny model gains the capability to alter the logits output by the LLMs. To make this tiny language model applicable to models with different vocabularies, we propose a novel token mapping strategy named PM-MinED. We have conducted extensive experiments on instruction tuning and unlearning tasks, demonstrating the effectiveness of CMC. Our code is available at https://github.com/wujwyi/CMC.

## 1 Introduction

In recent years, there has been an increasing number of large language models (LLMs) with varying parameter scales and vocabularies, whose outstanding performance has significantly impacted human society (Achiam et al., 2023; Zhao et al., 2023). At the same time, although large pre-trained language models have gained the ability to handle various natural language processing tasks after pre-training, they generally face a series of common optimization needs to meet specific application requirements or ethical standards. For instance, in the case of instruction following (Wei et al., 2022), after pre-training, vanilla models typically require instruction tuning to develop the capacity to comprehend user instructions accurately. Alternatively, unlearning and detoxification are also necessary considerations (Gehman et al., 2020; Chen and Yang, 2023). During the training process, models may encounter data containing real-world personal privacy information or content that is harmful, offensive, or prejudiced. This could lead them to output these information during the

---

[*]Corresponding Author

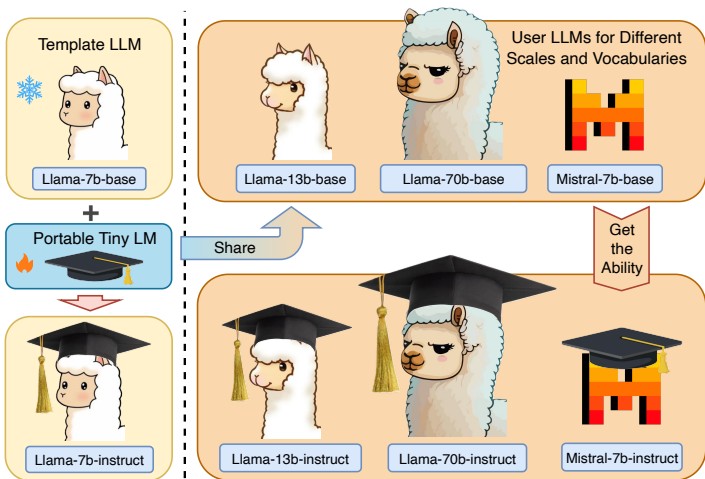

Figure 1: Cross-model control could apply the fine-tuning outcomes of one model to other models

inference stage (Wei et al., 2023; Huang et al., 2022). When deploying to a large number of users, it is crucial to avoid outputting these content.

However, existing methods usually could only optimize a single target model at a time, such as fine-tuning (Hu et al., 2022; Lester et al., 2021; Liu et al., 2024b), retraining (Keskar et al., 2019), or activation editing (Li et al., 2023a; Leong et al., 2023). These methods require altering the model's parameters or adding new parameters, which must align with the original parameters. These newly added or modified parameter values cannot be applied to models with different structures. Furthermore, although some guided decoding methods (Liu et al., 2021; Krause et al., 2021; Liu et al., 2024a) can be used for a few models of different scales within the same model family, they cannot be applied to models with other vocabularies. Moreover, these methods introduce a significant inference burden. For in-context learning (Dong et al., 2022), although this method can alter the behavior of models through natural language prompts, it falls short in satisfying the requirements of complex tasks such as instruction following or unlearning, which are better accomplished through fine-tuning. Consequently, a critical question arises: When model owners face constraints on data and computational resources, preventing them from directly fine-tuning their models, can they effectively leverage the fine-tuning outcomes of other LLMs at a lower cost to improve their models?

To solve this problem, we sought to explore the similarities in the fine-tuning effects across models with different parameter scales and vocabularies. We define the effect of fine-tuning as the change in the model's output logits after the fine-tuning process, compared to the logits before fine-tuning. We discovered that the shifts in logits across different models exhibit a high degree of similarity. It further inspires us to think: *Could a portable neural network model be utilized to alter the output logits of various models?* Thereby enabling a diverse range of models to achieve their optimization requirements through this neural network model.

In this paper, we propose Cross-model Control (CMC), a method that could improve multiple LLMs in one-time training with a portable tiny language model. As demonstrated in Figure 1, we introduce a tiny language model with significantly fewer parameters than mainstream LLMs. This model is trained alongside a frozen template LLM, enabling the tiny language model to alter the logits output by the LLM. Subsequently, to facilitate its application across models with different vocabularies, we introduce the strategy of prefix match with minimum edit distance (PM-MinED), a lightweight approach for aligning the vocabularies of the user LLM and the tiny language model at the token level. Through this approach, we achieve the training of a single model that concurrently improves multiple models. We conducted extensive experiments on instruction tuning and unlearning tasks, demonstrating the effectiveness of CMC, where a tiny language model with only 15 million parameters can empower a large model with 70 billion parameters, which is thousands of times larger. Our contributions can be summarized as follows:

- To the best of our knowledge, we are the first to propose a training method that improves multiple models in one-time training. This approach facilitates the multiple utilizations of

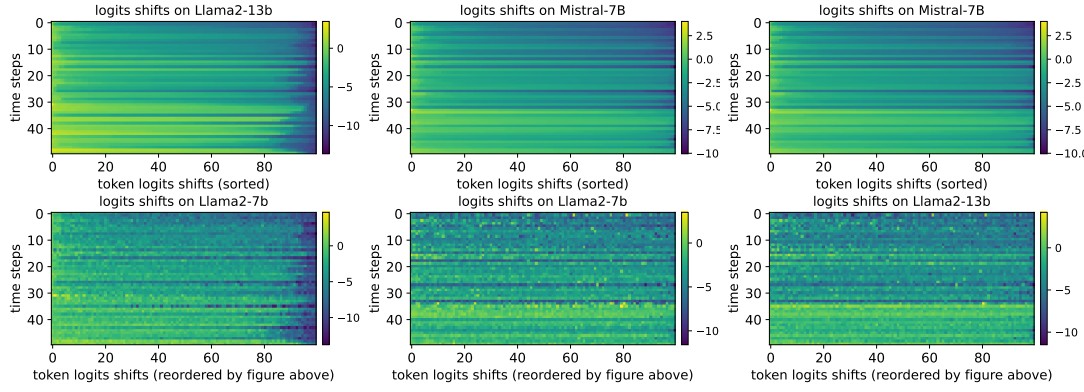

(a) Logits shifs on LLAMA2-13B and LLAMA2-7B.

(b) Logits shifs on MISTRAL-7B and LLAMA2-7B.

(c) Logits shifs on MISTRAL-7B and LLAMA2-13B.

Figure 2: Logits shifs on different models exhibit a high degree of similarity.

fine-tuning outcomes, enabling LLM owners who lack data and computational resources to improves their models. Showcasing a novel method for model enhancement.

- We conducted a detailed analysis of the similarities in fine-tuning across different models and discovered that the shifts in logits for the same task are similar across various models.

- Through extensive experimentation, we have demonstrated the effectiveness of our proposed method. Moreover, we discovered that language models with minimal parameter sizes possess significant potential in assisting LLMs.

## 2 Preliminaries

Even though different LLMs may have varying parameter sizes and distinct vocabularies, if these models are fine-tuned on the same task, does the effect of fine-tuning exhibit similarities across them?

Given a dataset $\mathcal{D}$ and an vanilla model $\mathcal{M}^v$, we fine-tune the model $\mathcal{M}^v$ using dataset $\mathcal{D}$ to obtain a fine-tuned model $\mathcal{M}^d$. Provided with a prompt, the models $\mathcal{M}^v$ and $\mathcal{M}^d$ predict the next token, resulting in $\zeta_v \in \mathbb{R}^{|\mathcal{V}|}$ and $\zeta_d \in \mathbb{R}^{|\mathcal{V}|}$ respectively, where $\mathcal{V}$ denotes the size of the vocabulary. We define the effect of fine-tuning as the as the shift in logits, that is $\zeta_d - \zeta_v$.

When comparing the effects of fine-tuning between two distinct models, $\mathcal{M}_1^v$ and $\mathcal{M}_2^v$, it involves comparing the shifts in logits when encodeing the same sequence. Specifically, given an input-response pair, denoted as $[x_1, x_2, ..., x_n, y_1, y_2, ..., y_m]$, we fed this pair into the models $\mathcal{M}_1^v$, $\mathcal{M}_1^d$, $\mathcal{M}_2^v$, and $\mathcal{M}_2^d$. We then record the logits from the final layer output of each model. The logits corresponding to the response part are extracted, yielding $\zeta_1^v \in \mathbb{R}^{m \times |\mathcal{V}_1|}$, $\zeta_1^d \in \mathbb{R}^{m \times |V_1|}$, $\zeta_2^v \in \mathbb{R}^{m \times |\mathcal{V}_2|}$, and $\zeta_2^d \in \mathbb{R}^{m \times |\mathcal{V}_2|}$. To reduce the effects of varying scales in the logits from different models, we attempted to apply the LogSoftmax operation to the logits, thereby transforming them into the same logarithmic probability space. By calculating the difference between the logits after and before fine-tuning, we obtain the fine-tuning effects.

$$\mathcal{T}_{\mathcal{M}1} = \text{LogSoftmax}(\zeta_1^d) - \text{LogSoftmax}(\zeta_1^v) \tag{1}$$

$$\mathcal{T}_{\mathcal{M}2} = \text{LogSoftmax}(\zeta_2^d) - \text{LogSoftmax}(\zeta_2^v) \tag{2}$$

We selected three vanilla models, LLAMA2-7B , LLAMA2-13B , and MISTRAL-7B , encompassing diverse parameter scales and vocabularies. To investigate the similarity of fine-tuning effects across different models on the same dataset, we fine-tuned them individually on the GPT4-Alpaca dataset (Peng et al., 2023). All models were fine-tuned using low-rank adaptation (Hu et al., 2022), with hyperparameters provided in Appendix A.

To facilitate an intuitive comparison of fine-tuning effects across different models, we visualized the shifts in logits before and after fine-tuning in the form of heat maps[2]. Taking Figure 2a as an example,

---

[2]We use "What causes the northern lights?" as the input, and the output of $\mathcal{M}_1^d$ as the response.

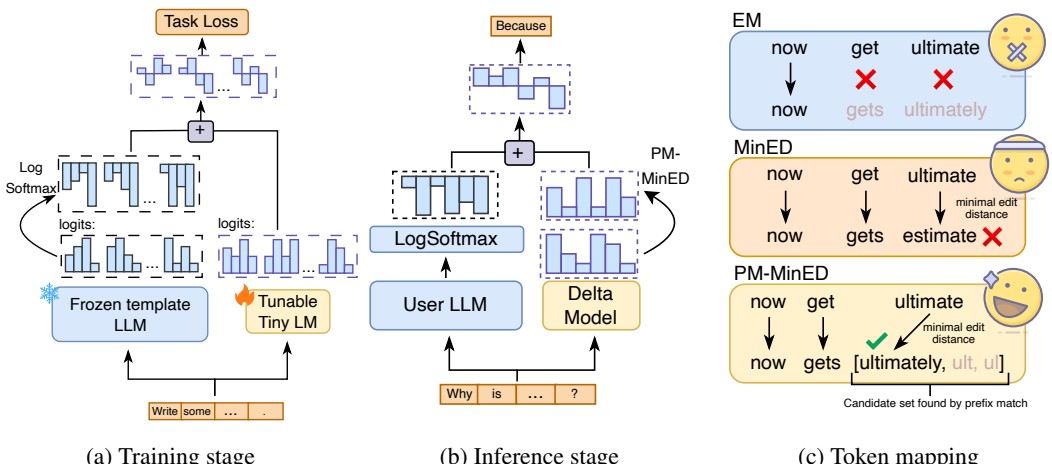

(a) Training stage      (b) Inference stage      (c) Token mapping

Figure 3: Overview of Cross-model Control

we selected the top 100 tokens with the highest logits values in $\zeta_{\text{LLAMA2-13B}}^{GPT4-Alpaca}$ at each time step, sorted according to the values of $\mathcal{T}_{\text{LLAMA2-13B}}$, and applied the corresponding indices to $\mathcal{T}_{\text{LLAMA2-7B}}$. We found that the upper and lower sub-figures in Figure 2 were highly similar, both exhibiting a trend of larger values on the left and smaller ones on the right, indicating a remarkably similar fine-tuning effect of different models on the same dataset. Additionally, we have quantitatively analyzed the fine-tuning effects across models using Sinkhorn divergence, as detailed in Appendix D.

## 3 Cross Model Control

Given the observation that the logits shifts before and after fine-tuning in different LLMs are remarkably similar, we introduce a parameter-efficient, portable tiny language model designed to learn to alter the output logits of LLMs, which we call the "delta model". Subsequently, we introduce a lightweight token mapping strategy called PM-MinED. Empowered by PM-MinED, the delta model becomes applicable to other LLMs with varying parameter sizes and vocabularies. An overview of the cross-model control is illustrated in Figure 3.

### 3.1 Training the portable delta model to alter the output of LLM

In Section 2, we found that the logits shifts before and after fine-tuning LLM are very similar. Therefore, we attempt to fit the logits shifts of LLM using a delta model. Specifically, as shown in figure 3a, we train the template LLM and delta model together, keeping the template LLM frozen and the delta model tunable. During forward propagation, we add the logits of LLM and the logits of the delta model together to obtain the final logits, aiming to teach the delta model to alter the logits of LLM. Considering that the logits of different LLM outputs have different scales, to enhance the delta model's applicability during the inference stage after training, we apply a LogSoftmax to the logits of LLM to project them into logarithmic space.

Formally, we denote the template LLM as $\mathcal{M}_t$, the delta model as $\mathcal{M}_d$, a training data sample as $x$, the logits output by $\mathcal{M}_t$ and $\mathcal{M}_d$ at the final layer as $\zeta_t$ and $\zeta_d$ respectively, and the final logits used for calculating the loss as $\zeta_{final}$.

$$\zeta_t = \mathcal{M}_t(x), \quad \zeta_d = \mathcal{M}_d(x) \tag{3}$$

$$\zeta_{final} = \text{LogSoftmax}(\zeta_t) + \zeta_d \tag{4}$$

We do not apply LogSoftmax to the logits $\zeta_d$ of the delta model because it would lead to poorer performance, which we will explain in Section 3.2.

### 3.2 Sharing the portable delta model with other LLMs

After the delta model has acquired the capability to alter LLM logits, we share it with other user LLMs to achieve fine-tuning effects. During the inference stage, the interaction between the LLM

and the delta model remains consistent with the training stage. We apply LogSoftmax to the logits output by the LLM, projecting them into logarithmic space, and then add them to the logits output by the delta model to obtain the final logits used for decoding. When applying the delta model to LLMs with different vocabularies, we employ a token mapping strategy called PM-MinED to map tokens from the delta model's vocabulary to tokens in the LLM's vocabulary. The implementation details of this strategy will be presented in Section 3.3.

Formally, we represent the user LLM as $\mathcal{M}_u$, a prompt as $[x_0, x_1, ..., x_{t-1}]$. At time step $t$, the logits output by $\mathcal{M}_u$ and $\mathcal{M}_d$ at the final layer are denoted as $\zeta_u^t$ and $\zeta_d^t$ respectively, where

$$\zeta_u^t = \mathcal{M}_u(x < t), \quad \zeta_d^t = \mathcal{M}_d(x < t) \tag{5}$$

The logits output by the user LLM assisted by the delta model at time step $t$ are represented as

$$\zeta_{final}^t = \text{LogSoftmax}(\zeta_u^t) + \alpha \cdot \text{TokenMap}(\zeta_d^t) \tag{6}$$

Here, TokenMap represents the PM-MinED strategy, and $\alpha$ denotes a strength coefficient that can adjust the intensity of alteration made by the delta model to the LLM outputs, which will be demonstrated in Section 4.4. We do not apply LogSoftmax to the delta model's logits in both Equations 4 and 6. This is because the outputs of LogSoftmax are all negative. If a token from the LLM cannot find its corresponding delta model token in the token mapping, the logits of this token will not be able to receive a negative adjustment.

## 3.3 Token Mapping Strategy PM-MinED

To enable the delta model to be adapted to user LLMs with varying vocabularies , it is imperative to establish a mapping relationship between the vocabularies of the user LLM and the delta model. In previous token mapping strategies, Fu et al. (2023) implemented an exact match method, which involves finding tokens in the delta model's vocabulary that are completely identical to the tokens of the user LLM; if a perfect match cannot be found, the matching attempt is abandoned. This method's limitation is that the logits of tokens that cannot be perfectly matched will not be adjusted by the delta model. Building upon this, Wan et al. (2024) introduced the minimum edit distance strategy, aiming to utilize tokens that cannot be perfectly matched by finding the token in the delta model's vocabulary with the smallest edit distance to the user LLM's token. However, this method might lead to matching tokens with the smallest edit distance but irrelevant semantics, such as erroneously matching "ultimate" with "estimate".

To overcome the aforementioned issues, we propose a new mapping strategy: Prefix-Match with Minimal Distance (PM-MinED). This strategy not only considers the edit distance between tokens but also introduces the concept of prefix matching to enhance the accuracy and semantic relevance of the mapping. As illustrated in Figure 3c, in the process of matching the token "ultimate" with the corresponding token in the delta model vocabulary, we initially create a candidate set consisting of tokens that either have "ultimate" as a prefix or are prefixes of "ultimate", which includes elements such as ["ultimately", "ult", "ul", "u"]. Subsequently, within this candidate set, by comparing the edit distances, the token "ultimately" is identified as the one most closely matching "ultimate".

# 4 Experiment

To evaluate the effectiveness of our method, we conducted experiments on two optimization tasks commonly required for LLMs: instruction tuning and unlearning, with the results presented in Sections 4.1 and 4.2. Furthermore, we analyzed the impact of different sizes of delta models and the strength coefficient $\alpha$ on performance in Sections 4.3 and 4.4. Finally, the outcomes of the ablation studies are demonstrated in Section 4.5, where we investigate the necessity of applying LogSoftmax to the logits of LLM outputs and employing prefix match during token mapping.

## 4.1 Experiment on Instruction Tuning

Instruction tuning refers to the process of fine-tuning pre-trained models to better understand and follow human natural language instructions.

**Training Dataset** We utilized the GPT4-Alpaca dataset (Peng et al., 2023) to train our delta model. This dataset consists of 52k instruction-following examples, with instructions sourced from Stanford Alpaca data (Taori et al., 2023) and responses generated by GPT-4.

Table 1: Instruction tuning results on AlapcaEval (Win %). In cross-model control, all base models incorporate the same delta model, which is trained using the LLAMA2-7B as the template model.

| Method | Params Add | LLAMA2-7B | LLAMA2-13B | LLAMA2-70B | MISTRAL-7B |
|---|---|---|---|---|---|
| Vanilla Base Model | - | 4.22 | 5.34 | 11.55 | 6.83 |
| LoRA (upper bound) | 110M | 68.18 | 75.16 | OOM | 79.91 |
| Proxy-tuning | 220M | 8.47 (+4.25) | 10.47 (+5.13) | 8.59 (-2,96) | - |
| CMC (ours) | 110M | 30.41 (+26.19) | 39.04 (+31.51) | 49.81 (+38.26) | 33.29 (+26.46) |

**Evaluation Method** We employed the AlpacaEval benchmark (Li et al., 2023b) to evaluate the instruction-following ability of our method. This benchmark includes 805 instructions, with GPT-4 serving as the annotator to compare the output of the tested model against Davinci003's output, using win rate as the evaluation metric.

**Implementation** We introduce TINYLLAMA-110M[3] as the delta model, which is based on the Llama (Touvron et al., 2023) architecture with a parameter size of 110M, featuring 12 Transformer decoder layers and a hidden size of 768. We use LLAMA2-7B as the template LLM to guide the delta model in learning to modify the output of the LLM. LLAMA2-13B, LLAMA2-70B , and MISTRAL-7B are selected as user LLMs to assess the effectiveness of the delta model on LLMs with different parameter sizes and vocabularies.

**Baseline Methods** **a) LoRA fine-tuning** (Hu et al., 2022): A parameter-efficient fine-tuning method that freezes the pre-trained model weights during training and incorporates trainable rank decomposition matrices into the Transformer layer to reduce the number of trainable parameters. As an immovable method, LoRA **is not suitable for direct comparison with our approach** but serves as a performance upper bound for reference. **b) Proxy-tuning** (Liu et al., 2024a): A method that does not directly fine-tune the model itself but selects a smaller-scale model within the model family as an anti-expert, fine-tunes it as an expert, and leverages the difference in logits between the expert and anti-expert during decoding for larger models. To facilitate a fair comparison, we use TINYLLAMA-110M as the anti-expert, fine-tuned on GPT4-Alpaca as the expert. This approach **cannot be directly applied to models with different vocabularies**.

**Analysis** The results of instruction tuning are shown in Table 1, where we observe that:

A single portable delta model can enable LLMs with different parameter sizes and vocabularies to achieve the ability to follow instructions. Furthermore, this ability is not constrained by the template model's capabilities, and as the user model's capabilities increase, the ability of the user model to follow instructions also increases. This indicates that the logits transformation for enabling a model to follow instructions can be applied to a wide range of LLMs, which is consistent with the findings in Section 2.

Our approach achieves better performance than Proxy-tuning with the same trainable parameters and fewer inference costs, as our delta model can alter the outputs of LLMs through training with the template model, while Proxy-tuning is constrained by the performance of the anti-expert model.

Our method's performance is not as good as LoRA's, as LoRA incorporates new tunable parameters in each Transformer layer, enabling deep interactions between the LoRA module and the model. However, this approach also prevents the LoRA module from being used as a portable neural network for models in different parameter spaces.

## 4.2 Experiment on Unlearning

Unlearning refers to the process of making a model forget specific information from the training data in order to prevent privacy leakage.

---

[3]https://huggingface.co/nickypro/tinyllama-110M

Table 2: Unlearning results on TOFU benchmark. All chat models incorporate the same delta model, which is trained using the LLAMA2-7B-TOFU as the template model. Better scores are bolded.

| Method | Forget Set | | | Retain Set | | | Real Author | | | World Fact | | |
|---|---|---|---|---|---|---|---|---|---|---|---|---|
| | RL ($\downarrow$) | P ($\downarrow$) | TR($\downarrow$) | RL($\uparrow$) | P($\uparrow$) | TR($\uparrow$) | RL | P | TR | RL | P | TR |
| LLAMA2-7B-TOFU | 0.99 | 0.99 | 0.51 | 0.98 | 0.99 | 0.48 | 0.93 | 0.45 | 0.58 | 0.87 | 0.43 | 0.56 |
| +LoRA | 0.01 | 0.00 | 0.77 | 0.71 | 0.75 | 0.48 | **0.88** | **0.51** | **0.68** | **0.88** | **0.46** | **0.60** |
| +CMC (ours) | **0.00** | **0.00** | **0.33** | **0.91** | **0.97** | **0.51** | 0.84 | 0.43 | 0.58 | 0.85 | 0.45 | 0.57 |
| LLAMA2-13B-TOFU | 1.00 | 1.00 | 0.46 | 0.99 | 1.00 | 0.53 | 0.89 | 0.51 | 0.67 | 0.86 | 0.46 | 0.62 |
| +$\delta$-UNLEARNING | 0.38 | 0.06 | 0.53 | 0.53 | 0.48 | 0.52 | 0.61 | 0.36 | 0.46 | 0.83 | 0.41 | 0.59 |
| +LoRA | 0.03 | 0.00 | 0.50 | 0.85 | 0.92 | 0.52 | **0.87** | **0.54** | **0.70** | **0.86** | **0.48** | 0.63 |
| +CMC (ours) | **0.00** | **0.00** | **0.29** | **0.97** | **0.99** | **0.55** | 0.77 | 0.49 | 0.65 | 0.83 | 0.47 | **0.65** |
| MISTRAL-7B-TOFU | 1.00 | 1.00 | 0.49 | 1.00 | 1.00 | 0.48 | 0.84 | 0.61 | 0.75 | 0.88 | 0.62 | 0.78 |
| +LoRA | **0.00** | **0.00** | 0.72 | 0.95 | 0.96 | 0.49 | **0.79** | 0.57 | 0.71 | **0.87** | **0.64** | **0.78** |
| +CMC (ours) | 0.00 | **0.00** | **0.30** | **0.99** | **0.99** | **0.51** | 0.73 | **0.62** | **0.75** | 0.86 | 0.63 | 0.77 |

**Evaluation Method** We utilized the TOFU benchmark (Maini et al., 2024) to evaluate our approach. The test data consists of four subsets of QA pairs, namely Forget Set, Retain Set, Real Author, and World Fact. The Forget Set and Retain Set contain fictitious author information, with the Forget Set representing the information to be forgotten and the Retain Set representing the information to be retained.[4] Real Author and World Fact are used to test the impact of unlearning on other knowledge within the model. Following TOFU's setting, we employed the following three evaluation metrics: **ROUGE-L** evaluates the matching degree between the output of the tested model and the ground truth answer; **Probability** assesses the conditional probability of the tested model outputting the correct answer; **Truth Ratio** calculates a ratio that compares the likelihood of its correct answer to an incorrect answer.

**Implementation** Initially, we trained LLAMA2-7B-CHAT , LLAMA2-13B-CHAT , and MISTRAL-7B-INSTRUCT on the Forget Set and Retain Set of TOFU to memorize fictitious author information, resulting in LLAMA2-7B-TOFU, LLAMA2-13B-TOFU, and MISTRAL-7B-TOFU. We selected TINYLLAMA-110M as the delta model, with the model learning fictitious author information, LLAMA2-7B-TOFU, serving as the template LLM, and LLAMA2-13B-TOFU and MISTRAL-7B-TOFU serving as user LLMs. During the training phase, we employed a gradient difference strategy, specifically conducting gradient ascent on the Forget Set and gradient descent on the Retain Set. The loss function can be expressed as:

$$\mathcal{L}_{diff} = -\mathcal{L}(S_F) + \mathcal{L}(S_R) \tag{7}$$

Here, $\mathcal{L}$ represents the cross-entropy loss, $S_F$ denotes the Forget Set, and $S_R$ denotes the Retain Set.

**Baseline methods** a) **LoRA fine-tuning**, which involves training each model separately. b) $\delta$-**UNLEARNING** (Huang et al., 2024) fine-tunes LLAMA2-7B-CHAT model, and guides LLAMA2-13B-CHAT during training and decoding to avoid outputting private information using the changes in logits between the tuned and the original LLAMA2-7B-CHAT. This method necessitates inferring three models during the inference, incurring substantial computational costs.

**Analysis** The results are shown in Table 2. The delta model trained with LLAMA2-7B-TOFU enables models with varying parameter scales and vocabularies to achieve unlearning effects. Its performance is comparable to LoRA while offering the capability of improving multiple models in a single training session. Furthermore, compared to $\delta$-UNLEARNING, our approach achieves superior performance at less than half the inference cost.

### 4.3 Impact of Parameter Size of Delta Model on Performance

We experimented with delta models of smaller parameter sizes, specifically the TINYLLAMA-42M[5] and TINYLLAMA-15M[6] models, and tested their performance on the first 50 data points of AlpacaEval.

---

[4]The test data for Forget Set and Retain Set are paraphrased and perturbed training data, serving as the ground truth answer and incorrect answer, respectively.

[5]https://huggingface.co/nickypro/tinyllama-42M

[6]https://huggingface.co/nickypro/tinyllama-15M

Table 3: Different delta model size on first 50 data points of AlpacaEval (win %).

| Delta Model Size | LLAMA2-7B | LLAMA2-13B | LLAMA2-70B | MISTRAL-7B |
|---|---|---|---|---|
| 15M | 40 | 50 | 72 | 52 |
| 42M | 54 | 64 | 82 | 50 |
| 110M | 54 | 66 | 74 | 54 |

The results are presented in Table 3. In most cases, we find that reducing the delta model's parameter size leads to a decrease in performance, but the performance does not rapidly decline as the parameter size decreases. When the delta model is applied to LLAMA2-70B, a larger delta model size may have a negative effect. We believe this is because the 70B LLM, with a massive parameter size, does not need to make significant adjustments to the output to acquire the ability to follow instructions, and an overly large parameter size in the delta model may lead to overfitting. This indicates that a delta model used for adjusting LLM output logits does not necessarily require a large number of parameters. It also demonstrates the significant potential for small models to assist large models.

## 4.4 Impact of Strength Coefficient on Performance

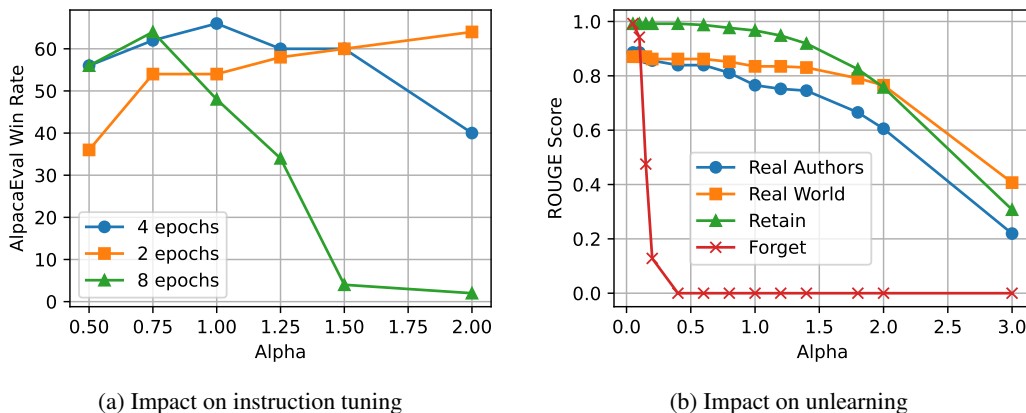

(a) Impact on instruction tuning        (b) Impact on unlearning

Figure 4: Impact of strength coefficient $\alpha$ on performance

We experimented with different strength coefficients $\alpha$ and observed their impact on performance.

For instruction tuning, we tested values within the range of [0.5, 2] and evaluated them on the first 50 data points of AlpacaEval. As shown in Figure 4a, we found that the performance was optimal when the $\alpha$ value was 1.0, and increasing or decreasing $\alpha$ resulted in decreased performance. This is similar to causing the delta model to overfit or underfit, so we explored whether adjusting the $\alpha$ value during inference could counteract the overfitting and underfitting during training. Specifically, the delta model performed best after training for four epochs, and we selected the checkpoint after training for two epochs as the underfitting scenario and the checkpoint after training for eight epochs as the overfitting scenario. We found that **adjusting the $\alpha$ value could counteract the overfitting or underfitting** during training. The underfitting delta model had a win rate of 54 when $\alpha$ was 1.0, which increased to 64 when $\alpha$ was 2.0. Similarly, the overfitting delta model had a win rate of 48 when $\alpha$ was 1.0, which increased to 64 when $\alpha$ was 0.75. This phenomenon indicates that our training method offers a considerable degree of fault tolerance, even if incorrect epoch parameters are set during the training phase, it is still possible to achieve the desired performance by adjusting the strength coefficient $\alpha$.

For unlearning, we found that **adjusting the value of $\alpha$ can serve as a balance between forgetting and retaining**. As shown in Figure 4b, increasing the value of $\alpha$ can prevent the model from outputting more sensitive information, but it may also lead to the loss of necessary information. Conversely, decreasing the value of $\alpha$ can allow the model to retain more essential information, but it may result in the model outputting more sensitive information. Users of LLM can flexibly adjust the $\alpha$ value based on the actual circumstances.

### 4.5 Ablation Study

In the ablation study, we examined the impact of not applying LogSoftmax to the logits output by the LLM and the effect of omitting prefix matching during token mapping on performance. As demonstrated in Table 4, our findings reveal that the removal of LogSoftmax results in a decline in performance. This suggests that projecting the logits output by the LLM into the logarithmic space can reduce the gap between training and inference. Similarly, the absence of prefix matching in token mapping diminishes the supportive effect of the delta model on the LLM. This indicates that prefix matching has a higher priority than selecting tokens with the minimal edit distance. Eliminating prefix matching could lead to the aggregation of token logits that are semantically unrelated.

Table 4: Ablation study

| Method | AlpacaEval (Win %) |
|---|---|
| LLAMA2-7B+delta model | 30.41 |
| w/o LogSoftmax | 28.64 |
| LLAMA2-13B+delta model | 39.04 |
| w/o LogSoftmax | 36.85 |
| MISTRAL-7B+delta model | 33.29 |
| w/o LogSoftmax | 31.22 |
| w/o Prefix Match | 30.19 |

## 5 Related Work

In this paper, we focus on modifying the outputs of the base language model. The related work can be broadly categorized into two approaches: training-intensive methods and decoding-time methods.

**Training-Intensive methods** One of the most effective methods for adapting large language models (LLMs) to specific downstream tasks involves updating only a subset of the model parameters, rather than the entire set. For instance, LoRA (Hu et al., 2022) achieves parameter updates by decomposing them into trainable low-rank vectors. Prefix-tuning (Li and Liang, 2021) introduces a series of continuous, task-specific vectors at the beginning of the input sequence for task adaptation. Adapter-tuning (Houlsby et al., 2019) integrates compact, trainable modules within the layers of a pre-trained model to facilitate transfer learning. BitFit (Zaken et al., 2021) selectively updates individual bias vectors in the model's parameters. Although these methods enhance performance in downstream tasks, they still require significant computational resources.

**Decoding-Time methods** To reduce training costs, some researchers have explored decoding-time methods. For example, DExperts (Liu et al., 2021) enhances or suppresses certain text attributes by integrating expert and anti-expert models during decoding. GeDi (Krause et al., 2021) employs smaller LMs as generative discriminators to steer the output of larger LMs, enhancing safety and control. Proxy-tuning (Liu et al., 2024a) modifies the predictions of an untuned larger model towards a desired outcome. ITI (Li et al., 2024) adjusts activation values during inference to generate more accurate responses. EFT (Mitchell et al., 2023) employs two smaller LMs to emulate the behavior of a larger LM without the need for additional training. Although these methods are effective, they often require the use of two additional language models during inference, increasing inference costs.

## 6 Conclusion and Limitation Discussion

In this paper, we introduced a training method named CMC, designed to improves multiple LLMs in one-time training, enabling LLM owners who lack data and computational resources to improve their models at a lower cost. Through comparative analysis of the logits shifts in different LLMs before and after fine-tuning, we observed a similarity in these shifts. Based on this observation, we introduced a portable tiny delta model to fit the logits shifts of LLMs, enabling the adjustment of outputs for LLMs of varying sizes and vocabularies. Our experiments in instruction tuning and unlearning tasks have demonstrated the effectiveness of our method.

However, our approach has certain limitations. The performance of the delta model is constrained by the scope of its vocabulary. If the vocabulary of the user's LLM contains tokens from languages not covered by the delta model's vocabulary, then the logits of these linguistically diverse tokens will not be adjusted accordingly.

## Acknowledgement

This work was supported by the National Natural Science Foundation of China under Grant No. 62377012 and the Special Fund for International Conferences of Graduate Students at East China Normal University.

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

# A Hyperparameters and experimental details

## A.1 Instruction Tuning

During the training of the delta model, we set the learning rate to 2e-4, batch size to 64, and trained for 4 epochs.

For LoRA fine-tuning, for both LLAMA2-7B and MISTRAL-7B models, we set r to 140 and alpha to 280, while for LLAMA2-13B, r is set to 90 and alpha to 180. For all models, the learning rate is 2e-5, batch size is 32, LoRA target are [q_proj,k_proj,k_proj], and the training lasted for 2 epochs.

When training the expert models for Proxy-tuning, the learning rate was set to 2e-4, batch size to 64, and trained for 16 epochs. We also experimented with varying the number of training epochs and found that performance was best with 16 epochs.

## A.2 Unlearning

In training the delta model, we set the learning rate to 1e-4, batch size to 16, and trained for 20 epochs.

For LoRA fine-tuning, across all models, r was set to 32, alpha to 64, with a learning rate of 1e-4, batch size of 16, and training spanned 5 epochs.

## A.3 Preliminaries

The hyperparameter settings for Section 2 were consistent with those outlined for instruction tuning.

# B Compute Resources

Our experiments were conducted on a server equipped with 512GB of memory and 4 Nvidia A100 40G GPUs.

# C Broader Impacts

Our approach can bring about positive social impacts. Specifically, our method allows for the reuse of the fine-tuning outcomes from one model to another. This attribute of supporting repeated use can reduce the cost of model training and decrease carbon emissions. Simultaneously, our method does not present any negative social impacts.

# D Quantitative Analysis of the Fine-tuning Effect

By calculating the difference between the logits after and before fine-tuning, we obtain the fine-tuning effects, and we evaluated their similarity by measuring the distance between them, which is assessed using Sinhorn divergence.

$$\text{Distance} = \text{Sinkhorn Divergence}(\mathcal{T}_{\mathcal{M}1}, \mathcal{T}_{\mathcal{M}_2})/|\mathcal{V}_1| \tag{8}$$

Table 5: Average Distance between logits shifts. Smaller distances mean more similarities

| Model1 | Model2 | Average Distance Between Logits Shifts | |
| --- | --- | --- | --- |
| | | Both train on GPT4-Alpaca | Model1 train on GPT4-Alpaca and Model2 train on GSM8k |
| LLAMA2-13B | LLAMA2-7B | 0.658 | 3.522 |
| MISTRAL-7B | LLAMA2-7B | 1.046 | 4.760 |
| MISTRAL-7B | LLAMA2-13B | 0.754 | 3.382 |

We fine-tuned them individually on the GPT4-Alpaca dataset. Additionally, to contrast the fine-tuning effects with other task, we also fine-tuned them on the GSM8k dataset (Cobbe et al., 2021). We selected the first 50 data from AlpacaEval as inputs, and used the output of $\mathcal{M}_1^d$ as the response to form input-response pairs. We chose the average Sinkhorn divergence as the indicator. The result is shown in Table 5. We observed that despite the models having different parameter scales and vocabularies, training on the same dataset still resulted in similar logits shifts. Conversely, significant differences were observed when training on different datasets.

