# OpenReview forum: "Cross-model Control: Improving Multiple Large Language Models in One-time Training"
_NeurIPS.cc/2024/Conference — NeurIPS 2024 poster_

### Official Review · Reviewer_aFGW · 2024-06-14

**Soundness:** 3
**Presentation:** 3
**Contribution:** 3
**Rating:** 6
**Confidence:** 4

**Summary:**

The development of the Cross-model Control method, which enables the use of fine-tuning outcomes from one model to improve other models through a portable tiny language model, effectively reducing training costs and computational resources.
The introduction of a novel token mapping strategy called prefix match with minimal edit distance (PM-MinED) to adapt the tiny model to different vocabularies of target LLMs.
Extensive experimental validation showing the effectiveness of CMC in common LLM optimization tasks like instruction tuning and unlearning, demonstrating substantial improvements over existing methods.
This approach represents a significant advancement in the field of LLM optimization, offering a scalable and resource-efficient solution for enhancing multiple models simultaneously.

**Strengths:**

The significance of this work lies in its potential impact on the practical deployment of LLMs. By enabling one-time, cost-effective optimization of multiple models, CMC addresses a critical bottleneck in the application of LLMs—resource intensity. This is particularly relevant in scenarios where computational resources are limited or where frequent re-training of models is not feasible. Additionally, the introduction of the PM-MinED strategy for token mapping broadens the applicability of the tiny delta model to diverse model architectures and vocabularies, thereby enhancing the utility of the proposed method in real-world applications.

**Weaknesses:**

The effectiveness of the Cross-model Control hinges significantly on the observed similarity in logit shifts across different models post-fine-tuning. This assumption may not hold in situations where models diverge significantly in architecture or training data, potentially limiting the applicability of the method in more diverse or adversarial settings.

**Questions:**

Generalizability Across Diverse Models:
How well does the Cross-model Control (CMC) method generalize to models with architectures or training procedures significantly different from those used in your experiments? For instance, how would CMC perform with emerging architectures like Transformer-XL or models trained with novel objectives?

**Limitations:**

The authors of the paper "Cross-model Control: Improving Multiple Large Language Models in One-time Training" have made an effort to address the limitations of their work, as well as discussing the broader societal impacts. However, there are areas where the discussion could be enhanced to provide a more comprehensive overview:
Scalability and Efficiency: The computational requirements and potential bottlenecks of implementing the PM-MinED strategy, especially across models with large and diverse vocabularies, are not thoroughly explored.
Bias and Fairness: There is a lack of discussion on how the method might influence or propagate biases within the trained models. Given the potential for large language models to perpetuate or amplify biases, this is a critical area that needs addressing.
Robustness and Error Analysis: The paper does not provide statistical robustness measures such as confidence intervals or error bars, which are essential for validating the reliability of the results.

Suggestions for Improvement
Expand on Computational Challenges: Include a detailed analysis of the computational efficiency and scalability of the PM-MinED strategy. Discuss potential strategies to optimize this process for larger-scale applications.

---

> ### Author Rebuttal · Authors · 2024-08-07
>
> We deeply value the time and effort you've invested in offering us constructive feedback. In response, we are endeavoring to address your points as follows:
>
> **Q1: Scalability and Efficiency: The computational requirements and potential bottlenecks of implementing the PM-MinED strategy, especially across models with large and diverse vocabularies, are not thoroughly explored.**
> A1: Thanks for your insightful comments.
> * Regarding the efficiency of PM-MinED:
> The PM-MinED strategy is highly efficient, with a time complexity of O(nm), where n is the size of the delta model's vocabulary, and m is the size of the user model's vocabulary.
> In our experiments, using an Intel Xeon Gold 6148 CPU, aligning the Llama and Mistral vocabularies (32000 tokens) using a single thread takes only about 3 minutes. Scaling up to vocabularies in the hundreds of thousands takes only a few tens of minutes. With multi-thread optimization, the calculation time can be reduced to just a few tens of seconds, significantly less than the time required to train LLMs. We will update the multi-thread implementation of PM-MinED in the code repository.
> It is worth noting that vocabulary mapping is an independent step prior to model inference and does not affect the model's inference time. Additionally, once vocabulary mapping is completed, it can be applied to different delta model models with the same vocabulary, making it a one-time process.
> * Regarding the scalability of PM-MinED:
> The PM-MinED strategy contains both prefix matching and minimum edit distance matching, which better guarantees the accuracy of token matching. In our experiments, there are 30% of the tokens in the mistral vocabulary that are absent from the llama vocabulary, yet PM-MinED still demonstrates good performance.
>     * Experiment on Instruction Tuning
>         |||
>         |-|:-:|
>         |**Model**|**AlpacaEval (Win %)**|
>         |Mistral-7B **(Mistral Vocab)**|6.83|
>         |Mistral-7B + Delta Model **(Llama Vocab)**|33.29|
>         |
>     * Experiment on Unlearning
>         ||||||
>         |-|:-:|:-:|:-:|:-:|
>         |**Model**|**Forget Set Prabability**| **Retrain Set Prabability**| **Real Author Prabability** | **Word Fact Prabability** |
>         |Mistral-7B-TOFU **(Mistral Vocab)**|1.00|1.00|0.61|0.62|
>         |Mistral-7B-TOFU+Delta Model **(Llama Vocab)**|0.00 **(lower is better)**|0.99|0.62|0.63|
>         |
>
>     We will explore the matching between vocabularies with greater differences in our future work.
>
> **Q2: Generalizability Across Diverse Models: How well does the Cross-model Control (CMC) method generalize to models with architectures or training procedures significantly different from those used in your experiments? For instance, how would CMC perform with emerging architectures like Transformer-XL or models trained with novel objectives?**
> A2: Thanks for your insightful comments. As the current mainstream LLMs adopt similar Transformer architectures and pre-training objectives[1], our work primarily focuses on the transfer of fine-tuning outcomes among transformer-based LLMs. We plan to explore the transfer of fine-tuning outcomes between more model architectures (such as Mamba and RWKV) in future work.
>
> References:
> [1] Zhao, Wayne Xin, et al. "A survey of large language models." arXiv preprint arXiv:2303.18223.
>
> **Q3: There is a lack of discussion on how the method might influence or propagate biases within the trained models. Given the potential for large language models to perpetuate or amplify biases, this is a critical area that needs addressing.**
> A3: Thanks for your insightful comments. In this study, our primary focus is on how to reuse fine-tuning outcomes to achieve the objective of improving multiple large language models through one-time training. In our experiments, we did not find that the delta model would amplify biases within the trained models. In future research, we will explore whether the CMC would have an impact on the biases of large language models.
>
> **Q4: Robustness and Error Analysis: The paper does not provide statistical robustness measures such as confidence intervals or error bars, which are essential for validating the reliability of the results.**
> A4: Thank you for your suggestion. The results of our experiments are derived from the average of multiple trials, and there is a significant performance improvement compared to the baseline. We will provide a more detailed error analysis in the next version. To facilitate reproducibility, we release the code repository and dataset available, and we will further release the trained models to the public.

---

> > ### Comment · Reviewer_aFGW · 2024-08-10
> > **Thanks for you response**
> >
> > I have thoroughly reviewed all the comments and the author’s responses, and I will remain postive of this submission.

---

> > > ### Author Response · Authors · 2024-08-12
> > >
> > > Thank you for your positive comments and we will accordingly revise our paper based on your suggestions.

---

### Official Review · Reviewer_V2UG · 2024-06-29

**Soundness:** 3
**Presentation:** 2
**Contribution:** 3
**Rating:** 5
**Confidence:** 3

**Summary:**

Large language models face a series of common optimization requirements under specific applications or ethical standards. Existing methods only optimize one target model at a time, which requires changing model parameters or adding new parameters. Authors found that the logit changes of different models before and after fine-tuning are very similar. Based on this finding, the CMC method introduces a small language model with a very small number of parameters, and trains it together with a frozen template LLM to enable the small model to learn to change the output logit of LLMs. In order to make the delta model suitable for models with different vocabularies, the paper proposes a novel token mapping strategy called PM-MinED, which is a lightweight method for aligning the vocabulary of user LLM and delta model at the token level.

**Strengths:**

1. The proposed method is novel and interesting. Unifying different training goals into one tiny model is interesting.
2. The proposed method has practical usage. It has significant potential to solve the problem of fintuning LLMs to fit different goals, especially for those users without enough resources.
3. The source code is given for reproducibility.
4. The experiment results show significant improvements of the proposed method.

**Weaknesses:**

1. Authors should provide more details about the hyper-parameter settings. Do the hyper-parameter is chosen as a suitable value with respect to different methods? The delta model is training with longer epochs than LoRA, which may make the delta model learn more than LoRA.
2. Figure 2 is not clear to me. It seems that the logits are sorted. So, how does this figure illustrate the similarity?
3. The Figure 3 is somehow misleading. Are output logits both fed into the user LLM and the Delta Model, or the raw training inputs are fed into the user LLM and the Delta Model?
4. It is under-explored that whether the Delta model harms the original ability of user LLMs.
5. The hardware configuration required to train and inference with the delta model is not provided.

**Questions:**

See weaknesses.

**Limitations:**

See weaknesses.

---

> ### Author Rebuttal · Authors · 2024-08-07
>
> We sincerely appreciate your valuable comments and feedback. We will describe more details of the experiments if more pages are provided.
>
> **Q1: "more hyper-parameter details" & "impacts of training epochs"**
> A1: **We selected the hyperparameters that achieve the best performance for the different methods.** In the instruction tuning experiment, LoRA and delta model performed best after 2 and 4 epochs respectively. Similarly, in the unlearning experiment, they performed best after 5 and 20 epochs respectively. The results of training LoRA for more epochs are shown in the table below. We observe that **training with longer epochs do not guarantee obtaining more or better performance**. One possible reason is that training for more epochs may cause the model to forget essential knowledge that should be remembered.
> - Training Mistral-7B with LoRA on the GPT4-Alpaca dataset and evaluating with the first 50 data points from AlpacaEval.
> |||
> -|:-:
> **Training Epochs**|**AlpacaEval (win %)**
> 1.00|94.00
> **2.00**|**96.00**
> 3.00|94.00
> 4.00|92.00
> |
> - Training and evaluating Mistral-7B-TOFU with LoRA on the TOFU benchmark.
> ||||||||
> -|:-:|:-:|:-:|:-:|:-:|:-:
> **Training Epochs**|**ROUGE Real Authors**|**Prob. Real Authors**|**Truth Ratio Real Authors**|**ROUGE Real World**|**Prob. Real World**|**Truth Ratio Real World**
> **5**|**0.79**|**0.57**|**0.71**|**0.87**|**0.64**|**0.78**
> 10|0.74|0.56|0.69|0.87|0.62|0.76
> 20|0.74|0.52|0.66|0.86|0.60|0.73
> |
>
> We'd like to provide more details in the supplementary material and updated version.
>
> **Q2: How does Figure 2 illustrate the similarity?**
> A2: As described in lines 97 to 100 of our paper, **in Figure 2, only the figure above is sorted according to the token logits shifts values. The figure below is reordered based on the token order of the figure above**. In other words, the tokens in the corresponding positions in the figures above and below are matched, and this matching is based on the PM-MinED strategy. This sorting is designed to facilitate the comparison of the similarities between the two figures. If the trends of color change from left to right in the figures above and below are similar, it indicates that the token logits shifts in the models of the two figures are similar, which suggests that the fine-tuning effects in the models of the two figures are similar. We will clarify it more clearly in the next version.
>
> **Q3: The input of the user LLM, template LLM and the delta model**
> A3: The training stage is represented by Figure 3a, where raw training inputs and responses are encoded by the tokenizer and fed to the template LLM and delta model. The inference stage is shown in Figure 3b, where the prompt and newly generated tokens are concatenated and then encoded separately by the user LLM and delta model's tokenizer before being fed to the user LLM and delta model, respectively. The output logits are not fed to any model. Instead, the output logits of the user model and delta model are added together to serve as the final logits for decoding and generating new tokens. We appreciate your thorough review and will include these details in the final version of our paper.
>
> **Q4: Whether the delta model harms the user LLM's original ability?**
> A4: In the experiments of unlearning and detoxifying, we explored whether the delta model would harm the original ability of user LLMs.
> - Exploration in the Unlearning Experiment
> As indicated in line 218 of the paper, "Real Author" and "Real World" represent the model's original knowledge. The impact of the delta model on these knowledge is presented in Table 2 of the paper, and we present the results again in the table below. Similar to LoRA, the delta model has a minimal impact on the model's original knowledge. Moreover, our proposed method achieved unlearning for multiple models in a single training session.
> ||||||||
> -|:-:|:-:|:-:|:-:|:-:|:-:
> **Method**|**ROUGE Real Authors**|**Prob. Real Authors**|**Truth Ratio Real Authors**|**ROUGE Real World**|**Prob. Real World**|**Truth Ratio Real World**
> Mistral-7B-TOFU|0.84|0.61|0.75|0.88|0.62|0.78
> +LoRA|0.79|0.57|0.71|0.87|0.64|0.78
> +CMC (ours)|0.73|0.62|0.75|0.86|0.63|0.77
> |
> * Exploration in the Detoxifying Experiment (New)
> We further explored whether the delta model would harm the original abilities of the user LLM in detoxifying tasks. We utilized HarmfulQA, a dataset comprising 1960 harmful questions covering 10 topics, and generated non-toxic responses using gpt-3.5-turbo to form the training dataset. We trained Nous-Hermes-Llama2-13b with LoRA as the baseline method, using Nous-Hermes-Llama2-7b as the template LLM to train the delta model. We evaluated the detoxification effects on the DangerousQA and AdversarialQA datasets and evaluated whether the model's original ability was compromised on the TruthfulQA dataset.
> The results, as shown in the table below, indicate that the delta model significantly reduced the likelihood of the model producing toxic outputs. Furthermore, similar to LoRA, **the delta model preserved the original ability of user LLM well**. We are committed to including related dicussion in the next version of our paper.
> |||||
> -|:-:|:-:|:-:
> **Method**|**DangerousQA (toxic %)**|**AdversarialQA (toxic %)**|**TruthfulQA MC (True %)**
> Nous-Hermes-Llama2-13b|0.445|0.897|0.430
> +LoRA|0.035|0.485|0.400
> +CMC (ours)|0.087|0.389|0.406
> |
>
> **Q5: Hardware configuration.**
> A5: As shown in the table below, the computational overhead associated with training and inference for the delta model is relatively small. Due to the small number of parameters of the delta model, the hardware requirements for inference with the delta model are approximately identical to inference the user LLM only.
> |||
> -|:-:
> **Model**|**Hardware Configuration**
> Train Stage|
> 7B/13B Models with LoRA|1 A100 40G GPU
> Delta Model 15M/42M/110M with Llama2-7b|1 A100 40G GPU
> Inference Stage|
> 7B/13B Models with LoRA/Delta Model|1 A100 40G GPU
> Llama2-70b with Delta Model|4 A100 40G GPUs
> |

---

> > ### Comment · Reviewer_V2UG · 2024-08-09
> > **Thanks for responses**
> >
> > Thanks for efforts in responses. After reading the response, I'd like to raise my score as 5.

---

> > > ### Author Response · Authors · 2024-08-12
> > >
> > > Thank you again for your valuable comments that help improve our paper!

---

### Official Review · Reviewer_VAGR · 2024-07-11

**Soundness:** 4
**Presentation:** 3
**Contribution:** 4
**Rating:** 8
**Confidence:** 5

**Summary:**

This paper proposes a novel training method named CMC, designed to improve performance for multiple LLMs in one-time training, thereby reducing training costs by reusing fine-tuning outcomes. The core approach of the paper consists of three steps:
1. Introducing a portable tiny language model (delta model) with a small number of parameters, trained alongside the frozen template LLM, to enable the delta model to alter the output logits of the LLM.
2. Introducing a novel token mapping method called PM-MinED, which aligns the vocabularies of the delta model and the user LLM.
3. Sharing the delta model with other user LLMs to simulate the effect of fine-tuning.
The author conducts extensive experiments on instruction tuning and unlearning tasks to demonstrate the effectiveness of the proposed method. Additionally, the paper highlights the significant potential of small language models in assisting large language models.

**Strengths:**

1. This paper is well-motivated. The author identifies the pain points of existing model optimization methods that are not portable, and provides a detailed analysis of the fine-tuning similarity between different models, effectively leading to the core method.
2. CMC is simple yet effective. The delta model serves as a parameter module that can be ported to models with different parameter scales and vocabularies, still achieves impressive performance. Its performance significantly surpasses baseline methods that can only be ported within the model family, and it achieves similar performance to the non-portable method LoRA in the unlearning task.
3. CMC has strong generalization, demonstrating good performance when applying delta models of different parameter scales to user LLMs with different parameter scales and vocabularies. It also exhibits good robustness, being able to effectively mitigate overfitting and underfitting phenomena through the adjustment of the strength coefficient.
4. This paper showcases the great potential of small models in assisting super large models, which is quite exciting!
5. The author has open-sourced the complete implementation process, which is easy to reproduce.

**Weaknesses:**

There are no apparent weaknesses, just some minor concerns.
1. In the vocabulary of language models, in addition to tokens that make up most of the words, there are still many special tokens. Mapping between special tokens from different vocabularies may require a significant amount of manual annotation.
2. The author needs to clarify how to represent a token that does not exist in the delta model's vocabulary when the user LLM generates it during inference. I'm concerned that if the token is directly encoded using the delta model's tokenizer, it may not adequately represent the information of that token.
3. The author would do well to include the pseudocode of the algorithm in the paper, which would clearly demonstrate the details of the inference part.

**Questions:**

1. As mentioned in the weaknesses, during the inference stage, if the user LLM generates a token that does not exist in the delta model's vocabulary, what is the input to the delta model when predicting the next token?
2. In the preliminary section, the author indicates that shifts in logits across different models are similar. Can we use a delta model to directly fit the logits shifts of existing models, such as fitting the logits shifts of Llama2-7b-chat and Llama2-7b-base? Will token-level logits shifts provide more fine-grained supervision signals, leading to better performance?

**Limitations:**

The authors have adequately addressed the limitations and risks of their work.

---

> ### Author Rebuttal · Authors · 2024-08-07
>
> We deeply value the time and effort you've invested in offering us encouraging feedback. In response, we are endeavoring to address your points as follows:
>
> **Q1：During the inference stage, if the user LLM generates a token that does not exist in the delta model's vocabulary, what is the input to the delta model when predicting the next token?**
> A1: Thanks for your insightful comments. When preparing inputs for the delta model, if the tokens generated in the previous step are not in the vocabulary of the delta model, we will not directly encode this single token using the delta model's tokenizer. Instead, we will first decode the input from the previous time step into natural language using the tokenizer of the user LLM, then we concatenate the newly generated token with the decoded sequence, and finally we re-encode the concatenated sequence using the tokenizer of the delta model. This approach helps to avoid directly encoding single tokens that are not in the vocabulary. Additionally, the tokenizer has a high encoding efficiency. Thus, there is almost no increase in the inference cost caused by this re-encoding process. We'd like to incorporate these details into the revision.
>
> **Q2: In the preliminary section, the author indicates that shifts in logits across different models are similar. Can we use a delta model to directly fit the logits shifts of existing models, such as fitting the logits shifts of Llama2-7b-chat and Llama2-7b-base? Will token-level logits shifts provide more fine-grained supervision signals, leading to better performance?**
> A2: Thank you for your new insights. In the early experimental phase, we have tried this method and attempted to measure the differences between the logits of the delta model and the logits shifts of Llama2-7b using methods such as KL divergence and Sinkhorn divergence. However, the loss during the training phase did not converge, and the evaluation results were also poor. We speculate that this may be due to the delta model's inability to handle such fine-grained information effectively.
>
> **Q3: In the vocabulary of language models, in addition to tokens that make up most of the words, there are still many special tokens. Mapping between special tokens from different vocabularies may require a significant amount of manual annotation.**
> A3: Thanks for your insightful comments. The special tokens that require manual alignment mainly include partial hexadecimal numbers, start symbols, end symbols, and line breaks. Their alignment is not complex, and the manual annotations are worthwhile for reusing the fine-tuning outcomes.
>
> **Q4: The author would do well to include the pseudocode of the algorithm in the paper, which would clearly demonstrate the details of the inference part.**
> A4: Thanks for your insightful suggestion. We will add a brief algorithmic description with more succinct mathematical notations to make the method more clear.

---

### Official Review · Reviewer_FZv7 · 2024-07-13

**Soundness:** 3
**Presentation:** 3
**Contribution:** 3
**Rating:** 7
**Confidence:** 4

**Summary:**

This paper investigates an interesting problem in fine-tuning LLMs: how to reuse the fine-tuning outcomes of one model for other models to reduce training costs. The authors provide an important empirical finding: different models exhibit highly similar logit shifts before and after the same fine-tuning process. Based on this, the authors propose Cross-model Control (CMC), a method that improves multiple LLMs in a single training session using a portable tiny language model. Specifically, the authors integrate a tiny language model with minimal parameters, training it alongside a fixed template LLM. This tiny model gains the ability to alter the output logits of the LLM, making it applicable to other models. Experiments demonstrates its effectiveness.

**Strengths:**

1. How to improve multiple LLMs in one-time finetuning is really an interesting topic for me.
2. The method is well-motivated, reasonable, and experimentally validated.
3. The code has been provided, which is excellent.

**Weaknesses:**

1. As the authors discuss in the limitations section, if the user’s LLM vocabulary includes symbols from languages not covered by the delta model’s vocabulary, the logits for these different language symbols will not be adjusted accordingly. While I don’t expect the authors to overcome this limitation, it does imply an underlying assumption in the proposed method. I would like the authors to explicitly state this assumption in the methods section.

2. Because the findings in Figure 2 are crucial for proposing the method, I would like to see not only the qualitative analysis provided in Figure 2 but also a quantitative analysis. This should include the similarity of logits shifts for different models fine-tuned on the same dataset, as well as the similarity of logits shifts for the same and different models fine-tuned on different datasets.

**Questions:**

See "Weaknesses".

**Limitations:**

The authors provide a discussion on limitations.

---

> ### Author Rebuttal · Authors · 2024-08-07
>
> We sincerely appreciate your recognition of our work and the valuable suggestions. We would like to address each of the questions as follows:
>
> **Q1: Because the findings in Figure 2 are crucial for proposing the method, I would like to see not only the qualitative analysis provided in Figure 2 but also a quantitative analysis. This should include the similarity of logits shifts for different models fine-tuned on the same dataset, as well as the similarity of logits shifts for the same and different models fine-tuned on different datasets.**
> A1: Thank you for your suggestion. As described in line 103 of the paper, the quantitative analysis is presented in Appendix D. We evaluated the fine-tuning effect similarity across different models by measuring the distance between them, which is assessed using Sinhorn divergence. A smaller distance indicates a higher level of similarity.
>
> $$\text{Distance} =\text{Sinkhorn Divergence}(T_{M1},T_{M2})/ |V_1|$$
>
> Here, $T_{M1}$ denotes the logits shifts of model1, $T_{M2}$ ​denotes the logits shifts of model2 after vocabulary mapping, and $|V_1|$ denotes the vocabulary size of model1.
> We fine-tuned the models individually on the GPT4-Alpaca dataset. In addition, to compare the fine-tuning effects with other tasks, we also fine-tuned them on the GSM8k dataset. The results are shown in the table below. We observe that despite the models having different parameter scales and vocabularies, training on the same dataset still result in similar logits shifts. Conversely, if the models are trained on datasets from different tasks, they will acquire distinct capabilities, which consequently results in substantial variations in their logits shifts.
> |||||
> -|:-:|:-:|:-:
> **Models**||**Average Distance Between Logits Shifts**||
> Model1|Model2|Both train on GPT4-Alpaca|Model1 train on GPT4-Alpaca and Model2 train on GSM8k
> Llama2-13b|Llama2-7b|0.658|3.522
> Mistral-7B|Llama2-7b|1.046|4.760
> Mistral-7B|Llama2-13b|0.754|3.382
> |
>
> **Q2: As the authors discuss in the limitations section, if the user’s LLM vocabulary includes symbols from languages not covered by the delta model’s vocabulary, the logits for these different language symbols will not be adjusted accordingly. While I don’t expect the authors to overcome this limitation, it does imply an underlying assumption in the proposed method. I would like the authors to explicitly state this assumption in the methods section.**
> A2: Thank you for your suggestion. We will incorporate this assumption into the method section. Despite this premise, within the language scope covered by the delta model's vocabulary, the delta model is able to deliver the capabilities across various LLMs in a portable manner. This marks the first successful implementation of improving multiple large language models in a single training session.

---

> > ### Comment · Reviewer_FZv7 · 2024-08-07
> >
> > Thank you for your response. I have no further questions. After reviewing all comments and the author’s corresponding replies, I remain positive about this submission. The strengths I highlighted in my review make this paper a solid contribution.

---

> > > ### Author Response · Authors · 2024-08-12
> > >
> > > We highly appreciate your efforts in providing us with valuable comments. Thank you so much!

---

> > > > ### Comment · Reviewer_FZv7 · 2024-08-12
> > > > **Score Update**
> > > >
> > > > I have decided to increase my score.

---

> > > > > ### Author Response · Authors · 2024-08-12
> > > > >
> > > > > Thank you, we greatly appreciate your attention and time once again!

---

### Official Review · Reviewer_VCcy · 2024-07-15

**Soundness:** 3
**Presentation:** 3
**Contribution:** 3
**Rating:** 6
**Confidence:** 4

**Summary:**

The paper introduces Cross-model Control (CMC), a novel method to improve multiple large language models (LLMs) in a single training session using a portable tiny language model. The core idea hinges on the observation that the logit shifts before and after fine-tuning are similar across different models. Leveraging this, CMC trains a tiny model alongside a frozen template LLM to adjust the logits output by LLMs. The paper also proposes a token mapping strategy, PM-MinED, to make the tiny model applicable to models with different vocabularies. Extensive experiments on instruction tuning and unlearning tasks demonstrate the effectiveness of CMC. The paper provides the code for their implementation.

**Strengths:**

Originality: The paper introduces a novel approach to improving multiple LLMs simultaneously, which addresses a significant challenge in the field. The use of a portable tiny model to adjust logits across different models is innovative.

Quality: The method is thoroughly tested on two important tasks—instruction tuning and unlearning. The experiments are well-designed and demonstrate the effectiveness of the approach. The analysis of logit shifts and the detailed comparison with baseline methods like LoRA and Proxy-tuning add to the robustness of the study.

Clarity: The paper is well-written with clear explanations of the proposed method, experimental setup, and results. The figures and tables effectively illustrate the findings, making the concepts easier to grasp.

Significance: The proposed method has substantial implications for the efficient fine-tuning of LLMs, especially for model owners with limited data and computational resources. The ability to apply fine-tuning outcomes across different models can lead to significant cost reductions and broader applicability of advanced LLMs.

**Weaknesses:**

Token Mapping Strategy: While PM-MinED is introduced to handle different vocabularies, its effectiveness might be limited when dealing with highly diverse or specialized vocabularies not covered in the training data. This could impact the generalizability of the method.

Performance on Larger Models: The performance of the tiny model on extremely large LLMs (70B llama 2 for example) can sometimes lead to overfitting, as indicated in Section 4.3. The paper acknowledges this but does not provide a concrete solution beyond adjusting the model size and strength coefficient.

Limited Error Analysis: The paper lacks a detailed error analysis, particularly concerning the cases where the method fails or underperforms compared to other approaches like LoRA. A deeper investigation into these scenarios could provide valuable insights for future improvements.

**Questions:**

Token Mapping Generalizability: Can you provide more details on how PM-MinED performs with highly specialized or domain-specific vocabularies? Have you tested the method on such datasets?

Handling Overfitting: Beyond adjusting the model size and strength coefficient, are there other strategies you considered to mitigate the overfitting of the tiny model when applied to larger LLMs?

Error Analysis: Can you provide more detailed error analysis or failure cases where CMC underperforms compared to other methods? This would help in understanding the limitations and potential areas for improvement.

**Limitations:**

The authors have addressed the limitations of their work adequately, highlighting the constraints related to the scope of the tiny model's vocabulary and its impact on performance. They also discuss the potential for overfitting when applying the tiny model to very large LLMs and the need to adjust model parameters carefully.

---

> ### Author Rebuttal · Authors · 2024-08-07
>
> We deeply appreciate your encouraging feedback regarding the originality and significance of our work. We are pleased to address your inquiries in detail as follows:
>
> **Q1: While PM-MinED is introduced to handle different vocabularies, its effectiveness might be limited when dealing with highly diverse or specialized vocabularies not covered in the training data. This could impact the generalizability of the method. Can you provide more details on how PM-MinED performs with highly specialized or domain-specific vocabularies? Have you tested the method on such datasets?**
> A1: Thanks for your insightful comments. PM-MinED is a training-free method that does not require training data. It conducts tokenizer alignment between differnt vocabularies via prefix matching and edit distance minimization. As shown in the tables below, in our experiments, there are 30% of the tokens in the mistral vocabulary that are absent from the llama vocabulary, yet PM-MinED still demonstrates good performance. We'd like to provide more details regarding the effectiveness of PM-MinED across various vocabularies in the updated version.
> * Experiment on Instruction Tuning
>     |||
>     |-|:-:|
>     |**Model**|**AlpacaEval (Win %)**|
>     |Mistral-7B **(Mistral Vocab)**|6.83|
>     |Mistral-7B + Delta Model **(Llama Vocab)**|33.29|
>     |
> * Experiment on Unlearning
>     ||||||
>     |-|:-:|:-:|:-:|:-:|
>     |**Model**|**Forget Set Prabability**|**Retrain Set Prabability**|**Real Author Prabability**|**Word Fact Prabability**|
>     |Mistral-7B-TOFU **(Mistral Vocab)**|1.00|1.00|0.61|0.62|
>     |Mistral-7B-TOFU+Delta Model **(Llama Vocab)**|0.00 **(lower is better)**|0.99|0.62|0.63|
>     |
>
> **Q2: Handling Overfitting: Beyond adjusting the model size and strength coefficient, are there other strategies you considered to mitigate the overfitting of the tiny model when applied to larger LLMs?**
> A2: Thanks for your careful reading. In the experiments of investigating the impact of delta model size on performance, we observe that using a larger delta model for extremely large user LLMs still yields good performance, but compared to a smaller delta model, there may be a slight overfitting-like phenomenon. To enhance the robustness of a larger delta model when applied to extremely large user LLMs, besides adjusting the delta model size, we can reduce the learning rate during training and make adjustments to the strength coefficient alpha (default value of 1.0) during the inference stage. Specific examples are shown in the table below, where the delta model is trained based on the Llama2-7b template model. By slightly lowering the learning rate from 2e-4 to 1.5e-4, the win rate of Llama2-70b with the delta model increases from 74% to 80%. And the performance on smaller user LLMs (e.g., Llama2-13b) can also be maintained via altering the alpha value (1.00 --> 1.05).
> ||||
> |--|:--:|:--:|
> |**Delta Model Learning Rate**|**AlpacaEval (Win %)**|**AlpacaEval (Win %)**|
> ||Llama2-70b+Delta Model|Llama2-13b+Delta Model|
> |2e-4|74 (alpha=1.00)|64 (alpha=1.00)|
> |1.5e-4|80 (alpha=1.00)|64 (alpha=1.05)|
> |
>
> **Q3: The paper lacks a detailed error analysis, particularly concerning the cases where the method fails or underperforms compared to other approaches like LoRA. A deeper investigation into these scenarios could provide valuable insights for future improvements. Can you provide more detailed error analysis or failure cases where CMC underperforms compared to other methods? This would help in understanding the limitations and potential areas for improvement.**
> A3: Thank you for your suggestion. We have collected some failure cases and will incorporate them into the revision. In some cases, LoRA outperforms CMC because it introduces new parameters at each layer, resulting in deeper interactions. However, these new parameters are strongly tied to the model parameters, bring the LoRA module an immovable characteristic that limits its use in models with different model structures. On the other hand, the delta model, as a portable neural network, can be reused across different models, enabling the improvement of multiple large language models in a single training session.

---

> ### Author Response · Authors · 2024-08-13
> **Kind Reminder on Feedback Response**
>
> Dear Reviewer VCcy,
>
> Thank you for your valuable feedback on our paper. We have carefully provided responses to your comments. Could you kindly let us know if our clarifications address your concerns?
>
> We appreciate your time and assistance.
>
> Best regards,
>
> Authors

---

### Decision · Program_Chairs · 2024-09-25

**Decision:**

Accept (poster)

**Comment:**

Based on the overall positive reviews and the authors' thorough responses addressing the reviewers' concerns, I recommend a weak accept for this paper. CMC presents a novel and promising approach to efficiently fine-tune multiple LLMs, demonstrating significant potential for reducing training costs.